# The cost and quality of life impact of glaucoma in Tanzania: An observational study

Ian Murdoch[1,2]*, Andrew F. Smith[3,4], Helen Baker[1], Bernadetha Shilio[5], Kazim Dhalla[6]

**1** Institute of Ophthalmology, University College, London, United Kingdom, **2** Moorfield's Eye Hospital, London, United Kingdom, **3** Department of Ophthalmology, King's College London, London, United Kingdom, **4** Medmetrics Inc., Ottawa, Canada, **5** Ministry of Health, Community Development, Gender, Elderly and Children, National Eye Care Program, Dodoma, Tanzania, East Africa, **6** Dr Agarwals Eye Hospital, Dar Es Salaam, Tanzania, East Africa

☯ These authors contributed equally to this work.
* i.murdoch@ucl.ac.uk

**Data Availability Statement:** The EQ-5D questionnaire is available from https://euroqol.org/wp-content/uploads/2019/10/Sample_UK__English__EQ-5D-5L_Paper_Self_complete.pdf. The VFQ-25 questionnaire is available from

## Abstract

### Aims

To determine the cost and quality of life impact imposed by glaucoma in Tanzania, East Africa.

### Methods

An expert panel of eye health professionals was convened to agree current glaucoma practice in Tanzania. In addition a structured patient survey was developed and administered. Supplemental cost and quality of life information was collected using cost questionnaires and validated quality of life measures, including the EQ5D and VFQ-25.

### Results

Key findings include following. Non-adherence is a major issue, especially in rural settings where over 50% of the patients may fail to return for review. Whilst medical therapy is overwhelmingly the first line treatment, the cost of maintaining this represents up to 25% of a patient's income. There is an impact of glaucoma on patients general well-being as determined by the EQ-5D and more tellingly on visual function with particular impact on role limitations as determined by the VF25. Despite our sample being taken in a private clinic and thus containing a much larger proportion of professionals than the general population, one third of the population earned Tanzanian Shillings (TZS) 170,000 per month which is below the minimum wage.

### Conclusion

These findings are of great importance for health care planners seeking to determine cost-effective, acceptable methods of both identifying and treating this major cause of preventable blindness.

https://www.rand.org/health-care/surveys_tools/vfq.html. All relevant data are within the paper and its Supporting Information files.

**Funding:** Author AFS is a consultant to Medmetrics Inc and was a paid health economics consultant on this project which was fully sponsored by The British Council for the Prevention of Blindness (BCPB) grant ID 172570 Mr Ian Murdoch. AFS was involved in the overall study design, data analysis and manuscript preparation phases of the entire project. The funder (BCPB) provided support in the form of salaries for authors [AFS] but did not have any additional role in the study design, data collection and analysis, decision to publish, or preparation of the manuscript. The specific roles of these authors are articulated in the 'author contributions' section.

**Competing interests:** Author AFS is a consultant to Medmetrics Inc and was a paid health economics consultant on this project. This does not alter our adherence to PLOS ONE policies on sharing data and materials. No other authors have any competing interests to declare.

## Introduction

Globally, it is estimated that the number of persons with both open angle glaucoma (OAG) and angle closure glaucoma (ACG) is rising dramatically. In 2010 Quigley et al estimated the worldwide prevalence of OAG and ACG combined was 60.5 million persons. This was projected to rise by 24% to 79.6 million persons by 2020. Similarly, the number bilaterally blind due to glaucoma was estimated to be 4.5 million for OAG and for 3.9 million for ACG in 2010 with a projected increase to 5.9 million for OAG and 5.3 million for ACG by 2020 [1]. In geographic terms Quigley et al showed that Sub-Saharan Africa (SSA) was the most adversely affected population globally with the highest ratio of glaucoma to adult population at 4.32% of the population in 2010. Such a disproportionate burden of glaucoma in SSA is further constrained by a lack of knowledge by the population as well as a lack of eye care resources and personnel. There is currently, for example, little understanding or consensus as to which glaucoma detection and treatment strategies are likely to be the most cost-effective in terms of the greatest number of sight years gained on a sustainable cost basis. Moreover, a recent literature review conducted by Smith et al (2018) determined that while a myriad of effective glaucoma control interventions exist in many urban centres throughout SSA, their widespread use and diffusion across SSA remains challenging principally due to their high cost and the lack of publicly funded eye health investments in glaucoma screening, detection and treatment alternatives [2]. This current situation across much of SSA exists despite the clear urgency of the problem at least as recognized by the global Vision 2020 coalition on blindness prevention and treatment [3]. In order to further inform the development of cost-effective strategies the aim of this project was to assess the patient and provider perceptions regarding glaucomas as well as their satisfaction with their specific therapy from both a cost and quality of life impact of primary open angle glaucoma.

## Methods

### Current practice and opinions of professionals

In addition to Ophthalmologists, Tanzania has a well-developed system of midlevel eye health workers who cater for the larger population that has no access to Ophthalmologists. The midlevel cadre consists of the Assistant Medical Officers (AMO-O), who specialize in clinical ophthalmology and cataract surgery; optometrists, ophthalmic nursing officers (ONO) and ophthalmic assistants. In the rural areas, eye clinics are mostly manned by AMO-Os, Ophthalmic nurses, Optometrists or any of the combinations. Patients are mostly seen by the midlevel cadre and, based on the competence or availability of facilities, are either managed at the same place or referred to a nearby regional hospital where an Ophthalmologist may be available. AMO-Os are trained to perform cataract surgery and emergency anterior segment surgeries like corneal lacerations. ONOs and optometrists are not trained to perform surgeries and therefore limit their services to medical treatment only. If resources at the regional hospital do not allow for appropriate treatment of the condition, a patient is referred to one of the 8 tertiary eye hospitals. There are several private eye hospitals with modern facilities in Tanzania, but they are beyond the reach of the majority.

In order to give an overall picture of the care provision on a countrywide/population basis a consensus meeting was convened in 2017. 22 health care professionals from throughout Tanzania attended in order to minimise bias. Fig 1 shows the distribution and profession of participants.

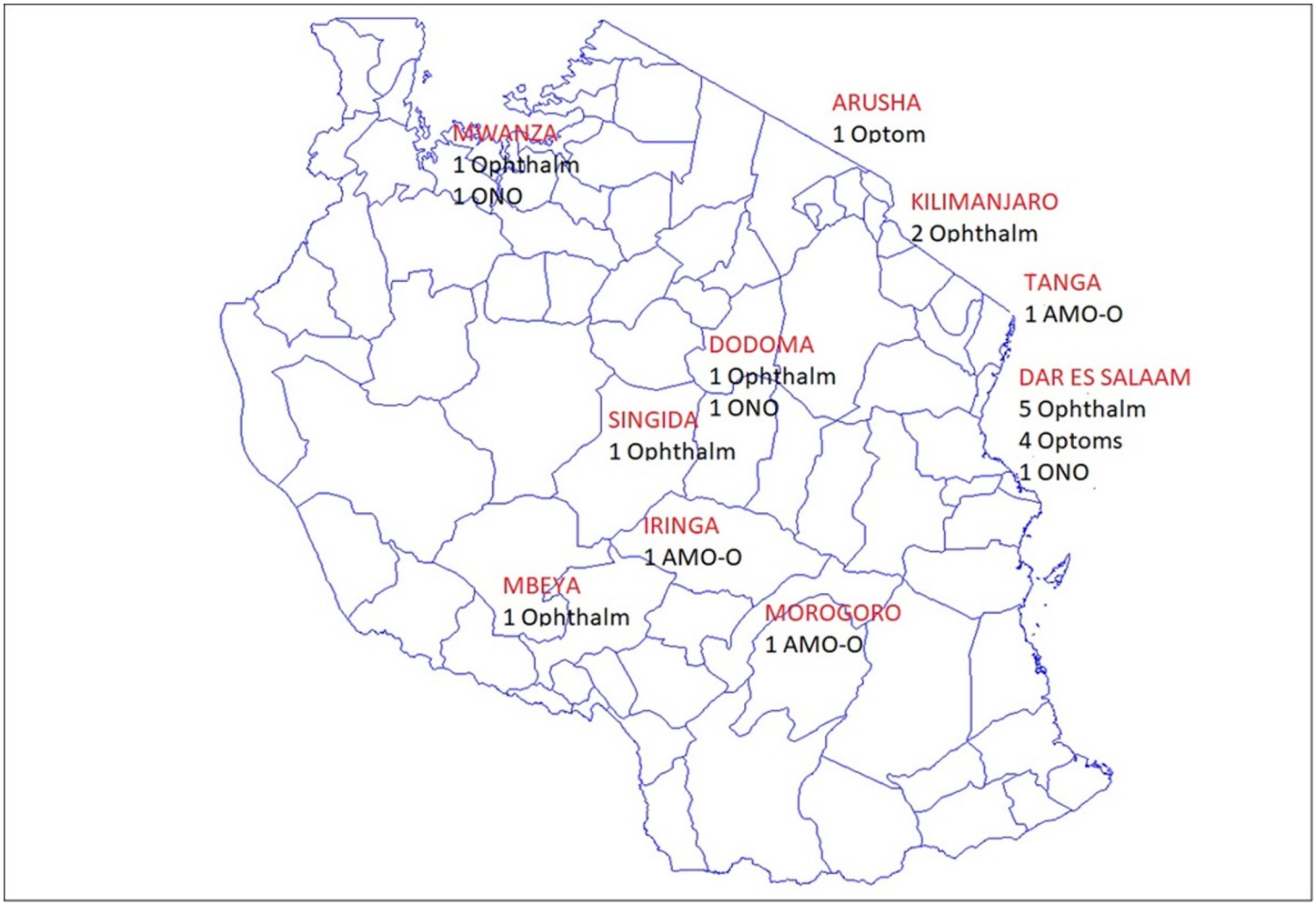

**Fig 1. Distribution of consensus meeting participants (The map is taken from wikipedia file: Tanzania Districts.png. (2014, November 25).** Wikimedia Commons, the free media repository. Retrieved 21.12, April 13, 2020 from https://commons.wiki,media.org/w/index.php?title = File:Tanzania_District.png&oldid = 140523089).

Opinions were sought on the proportions of glaucoma diagnoses attending, current management, patient adherence and attitudes, and concepts of 'ideal' practice in management of POAG. Finally, the content of a questionnaire to assess patient attitudes was agreed.

## Costs and QOL impact patient views

Ethical permission was granted by National Health Research Ethics Review Committee, National Institute for Medical Research, Tanzania ref NIMR/HQ/R.8a/Vol.IX/2683. Patients under active therapy attending Dr. Agarwal's Eye Hospital with Abu Baseer Specialist Eye clinic, Dar Es Salaam, Tanzania were invited to respond to three questionnaires delivered by two trained interviewers. All questionnaires were translated into Kiswahili and then back translated to ensure no loss of sense/meaning. All were delivered in Kiswahili. Convenience sampling of patients attending the glaucoma clinics were interviewed provided they gave free consent to participation and were capable understanding and responding to the questionnaires. Responses were on an anonymous basis with no personal identifiers or clinical information being collected.

The first questionnaire was the EQ5D-3L [4]. This is a validated method of measuring general patient utility and overall health status covering the domains of mobility, self-care, usual activities, discomfort and anxiety. Three response levels were used of no, moderate and extreme impact. In addition, the individuals indicated their perception of health on a Visual Analogue Scale where 0 was dead and 100 perfect health.

The second questionnaire was the VFQ25 [5]. This is a validated method of assessing self-perceived visual health status. Responses to each of the survey questions are converted to a numerical value between 0 and 100, and the responses aggregated to provide information on domains including, general health, visual health, ocular pain, near activities, distance activities, social functioning, mental health, role difficulties, dependency, driving, colour vision, peripheral vision and a composite score. A further six additional questions were selected for inclusion by the professionals in the consensus meeting. These were in the domains of near vision, distance vision [2], role limitations and well-being [2].

The third questionnaire was generated following discussion in the consensus meeting and related to glaucoma therapy and costs. This questionnaire was piloted and modified based on patient feedback before the final version was used. The key elements of this questionnaire related to age, gender, marital status, educational attainment, working status, average monthly income, principal occupation, payment for glaucoma drops, frequency of follow up visits, assistance with attending clinic visits, number of eye surgeries for glaucoma, frequency of eye drop usage, use of spectacles for near and distance vision, funding for treatment, travel times and travel costs associated with seeking treatment, waiting times for clinic visits, general impression with the glaucoma treatment received and overall willingness to pay for glaucoma surgery.

## Sample size and analysis

For the consensus meeting purposive sampling was used to ensure distribution of both levels and geographical location of care givers throughout Tanzania. For the patient questionnaires a target sample of a minimum of 118 patients was set in order to provide a margin of error of 9% with 95% confidence on questions with 50% response distribution [6]

Results of the consensus meeting and patient questionnaire were collated and presented using descriptive statistics. Analysis of the EQ5D and VF25 followed the methods adopted in prior literature in order to allow direct comparison [5,14].

## Results

### Current practice and opinions of professionals

**Current management.** POAG was identified as the single most common type of glaucoma presenting. All but one individual reported giving medical treatment as first line therapy to 95% or more patients. The remaining individual had a specialist practice and offered it to 85%. Reasons for medicine as first line therapy included the fact that for 12 of the attendees, medical treatment was the only available option. Many felt that offering surgery at an initial visit would intimidate patients returning to the centre and few centres offer this option. In addition, a fear of surgery was felt to exist in both patient and surgeons. Due to availability and affordability, timolol eye drops were the treatment of choice for a majority of participants.

Adherence to both prescribed therapy and repeat clinic appointments was a major issue. When asked the proportion of new patients that never returned the range was normally distributed from 2–60% with a mean of 18% and those only returning for 1 or two further visits a further 0–48% mean 19%. This varied hugely by practice and place, more rural settings having by far the highest proportion of non-adherents. Perceived reasons for not returning included

poor awareness of the disease, costs involved for no appreciable gain and alternative health or social priorities. Since many of the patients are elderly, they depend on others for socio-economic support.

The hierarchy of importance placed on health seeking behaviour was discussed. It was felt that social and occupational demands took precedence over health. The group thought symptomatic eye conditions such as cataract received much better attention by patients especially since, unlike glaucoma, the therapy resulted in tangible visual improvement. The issue of traditional and spiritual healing was discussed. It was generally felt that the lack of symptoms contributed to a belief by individuals that they had been cured by such therapies meaning they no longer attended clinics for treatment.

**Ideal management.** Optimal care pathways were discussed. Agreement was reached that a consultation time of 20 minutes for new and 15 minutes for follow-up was optimal with patients returning on average 3 or 4 times a year for review. Barriers to achieving this ideal were:

- Lack of resource, human, equipment, therapy and training

- Poor community/public awareness of the disease

- Poor health care provider motivation towards the disease with concentration on cataract, lack of perceived impact and poor training.

- Traditional and faith healing

## Costs and QOL impact patient views

**Demography.** All 129 participants in this study completed the three questionnaires. The demographic distribution of participants is shown in Table 1 together with the demographic composition for mainland Tanzania from the 2012 census [7] and living wage [8] for comparison. Our sample differs from the national demography in several important aspects. In keeping with an age-related disease, the majority (71%) were aged 65+ years. Our population also had considerably more individuals with higher levels of education and occupation than the national census.

**EQ5D.** Table 2 shows the main findings by domain. Over half (60%) reported no impact of their glaucoma on their health utility. The overall mean for the 5 EQ5D scores was 70 (range 18–94, SD 19).

The visual analogue scale (VAS) showed 60% of respondents to place themselves between 40–70% health, generally following a normal distribution. Only 6 (5%) individuals gave scores below 30. The VAS score ranged from 10–100 with a mean value of 65 (SD 18).

**VFQ-25.** Table 3 shows the findings for the various domains of the VFQ-25 as well as responses to nine additional questions. In general, the higher the VFQ25 score, the better the persons reported health outcome was, with 0 denoting the lowest possible score a patient could give, to 100 being the highest overall value. This grading applies to Question 1 through 16, while for question 17 through 25, the situation was reversed with 0 denoting the least impact and 100 denoting the greatest impact [5] Overall, patients report a mean VFQ25 score of 42 (SD = 17) for their overall health, with a score of 65 (SD = 16) for general overall visual health. Similarly, patients reported relatively little ocular pain with the mean score of 78 (SD = 24). Both near and distance activities were quite adversely impacted with VFQ25 scores of 57 (SD = 31) and 56 (SD = 32), respectively. Role limitations also yielded a reduced VFQ25 score of 52 (SD = 32). Peripheral, colour and driving vision were moderately impacted with VFQ25 scores of 65 (SD = 32), 80 (SD = 31) and 68 (SD = 29), respectively. With respect to the

**Table 1. Responses of glaucoma patients attending the clinic in Dar-es-Salaam concerning demographic details.**

| Variable | Options | Our sample N = 129 | National demography Tanzania mainland 2012 census (7) N = 24,454,247 |
|---|---|---|---|
| Age | 15/24 | 3 (2%) | 93% |
| | 25/34 | 2 (2%) | |
| | 35/44 | 7 (5%) | |
| | 45/54 | 14 (11%) | |
| | 55/64 | 25 (19%) | |
| | 65/74 | 46 (36%) | 7% |
| | 75+ | 32 (25%) | |
| Gender | Female | 50 (39%) | 51% |
| Highest education completed (denominator for percents 123 to match national data) | None | 6 (-) | |
| | Primary school | 38 (31%) | 83% |
| | Secondary school | 49 (40%) | 14% |
| | College | 6 (5%) | 1% |
| | University/Post graduate | 30 (24%) | 2% |
| Occupation | Farmer/manual laborer | 16 (12%) | 76% |
| | Semi-skilled manual | 16 (12%) | 9% |
| | Skilled tradesperson | 26 (20%) | 6% |
| | Professional | 43 (33%) | 3% |
| | Other | 28 (21%) | 5% |
| Monthly income by taxation bracket (TZS) | 0/169,999 | 44 (34%) | Living wage for individual (8) |
| | 170,000/359,999 | 25 (19%) | 211600–311000 |
| | 360,000/539,999 | 19 (15%) | |
| | 540,000/719,999 | 5 (4%) | |
| | 720,000+ | 36 (28%) | |

additional questions, these proved moderately problematic for patient. Role limitation, reading and seeing the TV were more affected.

**Patient questionnaire.** Individuals had travelled between 4 and 1500 Km to attend the appointment. About one third (36 (31%)) had travelled less than 10Km and another third (35 (31%)) had travelled between 10 and 100Km. 77 (60%) needed someone to come with them for their appointment. We extensively investigated associations with this need and found it weakly associated with increasing age ($Chi^2$ = 8.4 p = 0.2 using all age categories and $Chi^2$ = 4.1 p = 0.04 if age made binary by retirement age (<55 and 55+ years)). It was also weakly associated ($Chi^2$ 9.0 p = 0.06) with poor reported vision (VFQ-25 question 2). There was no

**Table 2. Findings of EQ5D questionnaire by domain in 129 individuals receiving therapy for glaucoma in an ophthalmology outpatient department in Tanzania.**

| EQ5D Domain | No Impact N (%) | Moderate Impact N (%) | Extreme Impact N (%) |
|---|---|---|---|
| Mobility | 71 (55) | 51 (40) | 7 (5) |
| Self-care | 111 (86) | 16 (12) | 2 (2) |
| Usual activities | 71 (55) | 33 (26) | 24 (19) |
| Pain/discomfort | 55 (43) | 69 (54) | 5 (4) |
| Anxiety/depression | 77 (55) | 39 (30) | 12 (10) |
| Total–Average % | 60% | 32% | 8% |

**Table 3. VFQ 25 Questionnaire findings in glaucoma patients attending the clinic in Dar-es-Salaam.**

|  | N | Mean | Std. Deviation |
|---|---|---|---|
| Overall health (Q.1) | 129 | 42 | 17 |
| Eyesight with correction both eyes Re Calculate (Q.2) | 129 | 66 | 16 |
| Ocular pain (Q.4,19) | 129 | 78 | 24 |
| Near activities (Q.5,6,7) | 119 | 57 | 31 |
| Distance Activities (Q.Q.8,9,14) | 91 | 56 | 32 |
| Social Functioning (Q.11,13) | 122 | 71 | 33 |
| Mental Health (Q.3,21,22,25) | 129 | 66 | 28 |
| Role Difficulties (Q.17,18) | 129 | 52 | 32 |
| Dependency (Q.20,23,24) | 129 | 69 | 33 |
| Driving (Q.15c, 16,16a) | 31 | 68 | 29 |
| Colour Vision (Q.12) | 125 | 80 | 31 |
| Peripheral Vision (Q.10) | 129 | 65 | 32 |
| Near vision trouble reading (Q. A1) | 129 | 65 | 32 |
| Difficulty recognising across room (Q.A6) | 128 | 83 | 24 |
| Difficulty seeing TV (Q.A8) | 129 | 70 | 32 |
| Role limitations (Q.A11 a and b) | 128 | 62 | 32 |
| Distress (Q.A12 and QA13) | 129 | 82 | 30 |

association with income, education, distance travelled, overall feeling of wellbeing, occupation or their perception of the care they were receiving at the clinic. Other responses are shown in Table 4.

Fifty-seven of participants (44%) had had at least one operation for glaucoma. This was not related to age, gender or monthly income. Time to travel to appointment was directly related to distance travelled showing internal consistency.

The median estimated total cost to attend for a glaucoma appointment was TZS 60000, at current exchange rates this is equivalent to £21.25. The median estimated cost of glaucoma drops per month was TZS 50,000. This was paid for by the patient themselves in 44% of cases (57/129) with a further 16% (20/129) covering some of the cost with assistance. The remainder were funded by alternative means including health insurance (45/129) or not on medication (7/129).

In order to make some comparison by individual between the cost of therapy per month and the reported monthly wage the monthly wage was taken as the mean of the range reported thus for each of the 25 individuals in the reported range TZS 170,000/359,999 the wage was placed as TZS 265000. The reported amount spent per month on drops was then expressed as a percentage of this amount. The results of this are shown in Fig 2 with a further division illustrating in which way this was funded. It can be seen the most common method of self-pay took up to 25% of the monthly salary.

## Discussion

Our patient population differs from the national figures in several important aspects. It is not surprising that the age range is older since primary open angle glaucoma is strongly age related. The predominance of males in the patient population represents a gender bias in access to treatment as noted in other studies in Africa [9].

Our study was undertaken in a private eye hospital where all patients pay. In addition, this is in an urban setting. The hospital setting may explain the disproportionate number of individuals with higher income, occupational and educational status. None-the-less over one third

**Table 4. Responses of glaucoma patients attending the clinic in Dar-es-Salaam relating to the costs associated with on-going therapy for glaucoma.**

| Question | Options | Finding |
|---|---|---|
| Do you need someone with you for a glaucoma checkup? N = 129 | Yes | 77 (60%) |
| Do you wear spectacles? N = 129 | Distance | 59 (46%) |
| | Near | 76 (59%) |
| How long have you been on drops for glaucoma? N = 129 | No drops | 3 (2%) |
| | <6 months | 20 (16%) |
| | 6>24 months | 32 (25%) |
| | 24>60 months | 31 (24%) |
| | >60 months | 43 (33%) |
| How often do you have a glaucoma check-up? N = 129 | Once a year | 20 (16%) |
| | 2–3 year$^{-1}$ | 54 (42%) |
| | 4–5 year$^{-1}$ | 29 (22%) |
| | 6+ year$^{-1}$ | 26 (20%) |
| How much are you paying for drops each month? TZS N = 124 | Mean (SD, range) | 64782 (57813, 0–260000) |
| | Median (quartiles) | 50000 (22250–82500) |
| How much does it cost you to come to an appointment N = 126 | Median (quartiles) | 60000 (43000–100000) |
| How long does it take you to travel to this appointment from the place where you are staying N = 128 | Up to 2 hours | 84 (of whom 9 living further away and staying locally) |
| | Over 2 hours | 44 (time up to 48 hours) |
| How do you pay for your appointments? N = 126 | Self | 58 (46%) |
| | Self + | 20 (16%) |
| | Other | 48 (38%) |
| How do you pay for your medical treatment? N = 122 (4 not on therapy) | Self | 57 (47%) |
| | Self + | 20 (16%) |
| | Other | 45 (37%) |
| How do you pay for your surgery? N = 57 (had surgery) | Self | 26 (46%) |
| | Self + | 4 (7%) |
| | Other | 27 (47%) |

of patients reported a monthly wage below the lower limit of the national living wage. Patients are required to pay consultation charges at every visit, however, should the patient express his/her inability to pay part or the whole amount at every visit then the counsellor takes the matter to the hospital manager who after considering the case can either waive off the consultation charges totally for the following visit or gives appropriate discount.

It has been shown that patients with lower educational status have poorer [10] and those with higher education status better health seeking behaviour [11]. Lower educational status individuals are also less likely to be aware of the disease process and treatment options [12].

The affordability of therapy has not been addressed in prior literature. Despite the bias towards professionals in our population, nearly 2/3rds of the patients earned less than TZS 540,000 a month (US dollars 235 at conversion rate of 1 USD = 2295) and one third a maximum of 170,000 TZS per month (USD 74) The mean cost of glaucoma drops per month as reported by the patients was TZS 64,782 (median 60,000). This is about 12% of the salary of the middle wage earners and more than a third of the salary of the minimum wage earners. This excludes other costs involved in the treatment. This important fact highlights the need for

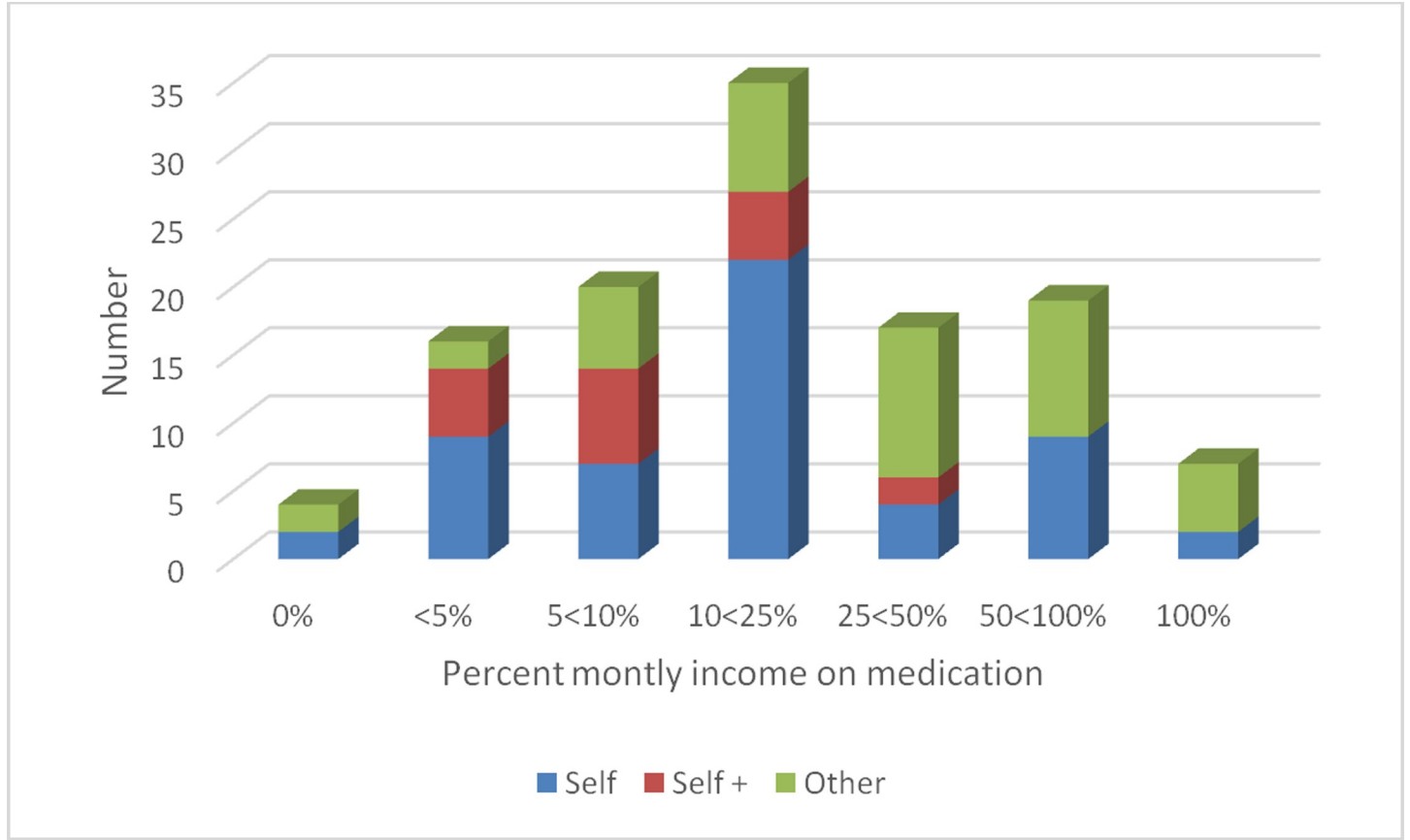

**Fig 2. Responses of glaucoma patients attending the clinic in Dar-es-Salaam concerning monthly medication cost expressed as a percentage of monthly salary.** Each bar chart has the number of individuals colour coded by the method by which the medications are funded.

clinicians to explore this social aspect of the patient before prescribing and planning follow up visits of their patients to minimize the cost burden thereby increasing compliance. It is also important for the health policy makers to know this in the hope that they may consider offering some subsidy to the glaucoma patients on expensive medicines. This fact may tip the balance towards one time surgical intervention rather than life-long medication.

One interviewer gave an anecdotal impression of the feelings of a majority of the people they interviewed and the current situation. They felt the overwhelming burden for patients was economic. Patients have poor understanding around glaucoma, how it can be treated and why they have it. There is a disinterest in therapy when told there isn't a cure and preventing progression is the only course of action. The lack of trained clinicians and insufficient resource is a big issue when managing and treating the large number of patients. Education and resource improvement were seen to be a key factor in future development of glaucoma management.

## EQ5D and VFQ25

Questions relating to subjective opinions of wellbeing are inevitably prone to huge variability. Not only does an individual's perception of health vary (some regard themselves relative to prior severe illness whilst others have never experienced serious illness for example) but their perception may differ from day to day according to mood/events/social situation etc. The

single question on the EQ5 using a visual analogue scale for general wellbeing showed an average of score of 65. The same question at the start of the VFQ25 was using words in 5 bins from excellent to poor and, when analysed by the accepted method, gave and average score of 42. We suggest this lack of internal consistency may be for two reasons. Firstly the EQ5D score ranges from best to worst imaginable health state whereas the scale in VF25 ranges from excellent to poor. We suggest that 'worst imaginable' is equivalent of 'very poor'. If the analysis is repeated using scores for the VF25 including a final category of very poor then the average score rises to 54. A second reason may be that people simply think in different ways when given a numeric scale and a verbal scale. This is strongly suggested by the fact that a majority considered their health good (71 (55%)) or fair (45 (35%)). Those who rated their health 'good' scored their health on the Likert scale 50 (14(20%)), 70 (14(20%)), 90 (13 (18%)). For the last of these scoring 90 there is little room for 'very good' and 'excellent'

The paucity of data from a comparable Sub-Saharan African population means direct comparison of our findings to EQ5D and VFQ25 measures for glaucoma patients is not possible. It is possible, however, to compare and contrast our findings to values obtained outside of SSA in other jurisdiction, albeit with the caveat that such comparison may not adequately reflect the provision of glaucoma services, nor the profile of glaucoma patients. To this end, various authors have reported a mean Visual Analogue Scale of ranging from 65 to 92 where 0 equals the worse possible outcome and 100 the best possible visual health status [13]. Similarly, Montemayor et al showed that the mean EQ5D scores among glaucoma patient in the Canadian Province of British Columbia was 81 (SD = 14) [14]. Aspinall et found that the mean value of the EQ-5D index among glaucoma patients in a clinic setting in Scotland was 76 (SD = 19) and that in patients with mild visual field loss, the mean EQ-5D index was 84 (SD = 17); in moderate visual field loss, 68 (SD = 21); and in severe visual field loss, 64 (SD = 26) [15]. More recently still, McDonald et al found that the mean EQ5D score in newly diagnosed POAG patients in England was 77 (SD = 22) [16]. Thus the EQ5D value obtained for our patients in Tanzania was at the lower range of EQ5D scores reported for the impact of glaucoma on patients. Our patients in Tanzania were much more likely to have moderate to severe perceived visual field loss associated which lower EQ5D scores, as found by Aspinall et al. [15] Whilst we did not collect clinical information, many papers have reported more severe disease in the clinic population in SSA compared to elsewhere [17].

In conclusion this study has the weaknesses discussed above of convenience sampling in an urban private clinic and use of subjective patient responses prone to lack of internal consistency. None-the-less the study has highlighted several key aspects relating to the current provision of care for glaucoma in SSA. A few are summarised below.

- Non-adherence is a major issue, especially in rural settings where over 50% of the patients may fail to return for review.

- Whilst medical therapy is overwhelmingly the first line treatment, the cost of maintaining this represents up to 25% of a patient's income which may tip the balance towards one-time costs such as surgical intervention

- There is an impact of glaucoma on patients general well-being as determined by the EQ-5D and more tellingly on visual function with particular impact on role limitations as determined by the VF25

- Despite our sample being taken in a private clinic and thus containing a much larger proportion of professionals than the general population, one third of the population earned <TzS170,000 per month which is below the minimum wage.

These findings are of great importance for health care planners seeking to determine cost-effective, acceptable methods of both identifying and treating this major cause of preventable blindness. 3293

## Supporting information

**S1 Data. STROBE statement—checklist of items that should be included in reports of observational studies.**
(DOC)

## Acknowledgments

We are grateful to Imani Kapesa and Emilika Kapesa for completing the patient interviews so expertly and to Imani Kapesa for data entry and oversight of the data integrity. The participants of the consensus workshop were Dr. Nkundwe Mwakyusa, Dr. Cyprian Ntomoka, Dr. Neema Daniel, Dr. Hassan G Hassan, Dr. Kazim Dhalla, Dr. Fariji Kilewa, Dr. Kuzenza Lufunga, Dr. Dilawar Padhani, Dr. Hussein Dattoo, Mr. Mudassir Alloo, Sr. Monica . . .., Sr. Emelda Lwena, Dr. Honest Maro, Dr. Japhet Boni, Mr. Akil Lalji, Dr. Christopher Mwanansao, Dr. Rajabu Kitumba, Dr. Secondri Njau, Dr. Frankie Sandi, Dr. Abdallah Dickemla, Dr. Frida Kassiane and Sr. Regina Mawalla. We wish to sincerely thank them for their willingness to travel and so constructively participate in this critical aspect of the project.

## Author Contributions

**Conceptualization:** Ian Murdoch, Andrew F. Smith, Helen Baker, Kazim Dhalla.

**Data curation:** Ian Murdoch, Helen Baker, Bernadetha Shilio, Kazim Dhalla.

**Formal analysis:** Ian Murdoch, Andrew F. Smith, Kazim Dhalla.

**Funding acquisition:** Ian Murdoch, Helen Baker, Kazim Dhalla.

**Investigation:** Ian Murdoch, Andrew F. Smith, Kazim Dhalla.

**Methodology:** Ian Murdoch, Andrew F. Smith, Helen Baker, Kazim Dhalla.

**Project administration:** Ian Murdoch, Kazim Dhalla.

**Resources:** Ian Murdoch, Bernadetha Shilio, Kazim Dhalla.

**Software:** Ian Murdoch.

**Supervision:** Ian Murdoch, Andrew F. Smith, Kazim Dhalla.

**Validation:** Ian Murdoch, Helen Baker.

**Visualization:** Ian Murdoch, Helen Baker.

**Writing – original draft:** Ian Murdoch, Andrew F. Smith, Helen Baker, Kazim Dhalla.

**Writing – review & editing:** Ian Murdoch, Andrew F. Smith, Helen Baker, Bernadetha Shilio, Kazim Dhalla.

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
