## [Decision Letter · Decision Letter 0]

1 Apr 2020

PONE-D-19-34308

The cost and quality of life impact of glaucoma in Tanzania: an observational study

PLOS ONE

Dear Mr Murdoch,

Thank you for submitting your manuscript to PLOS ONE. After careful consideration, we feel that it has merit but does not fully meet PLOS ONE’s publication criteria as it currently stands. Therefore, we invite you to submit a revised version of the manuscript that addresses the points raised during the review process.

We would appreciate receiving your revised manuscript by May 16 2020 11:59PM. To enhance the reproducibility of your results, we recommend that if applicable you deposit your laboratory protocols in protocols.io, where a protocol can be assigned its own identifier (DOI) such that it can be cited independently in the future. For instructions see: http://journals.plos.org/plosone/s/submission-guidelines#loc-laboratory-protocols

We look forward to receiving your revised manuscript.

Kind regards,

Prof Remco PH Peters, MD, PhD, DLSHTM

Academic Editor

PLOS ONE

Journal Requirements:

2. For reproducibility, please ensure you have included a reference to the survey questions given to ophthalmologists, or if they are not published, include a copy of the questions in manuscript or supporting information.

4. We note that Figure 1 in your submission contain map images which may be copyrighted. All PLOS content is published under the Creative Commons Attribution License (CC BY 4.0), which means that the manuscript, images, and Supporting Information files will be freely available online, and any third party is permitted to access, download, copy, distribute, and use these materials in any way, even commercially, with proper attribution. For these reasons, we cannot publish previously copyrighted maps or satellite images created using proprietary data, such as Google software (Google Maps, Street View, and Earth). For more information, see our copyright guidelines: http://journals.plos.org/plosone/s/licenses-and-copyright.

a)    You may seek permission from the original copyright holder of Figure 1 to publish the content specifically under the CC BY 4.0 license.  

5. Thank you for stating the following in the Financial Disclosure section:

We note that one or more of the authors are employed by a commercial company: Medmetrics Inc.

Reviewers' comments:

Reviewer's Responses to Questions

**Comments to the Author**

1. Is the manuscript technically sound, and do the data support the conclusions?

Reviewer #1: Yes

Reviewer #2: Yes

2. Has the statistical analysis been performed appropriately and rigorously? 

Reviewer #1: Yes

Reviewer #2: Yes

3. Have the authors made all data underlying the findings in their manuscript fully available?

Reviewer #1: Yes

Reviewer #2: Yes

4. Is the manuscript presented in an intelligible fashion and written in standard English?

Reviewer #1: Yes

Reviewer #2: Yes

5. Review Comments to the Author

Reviewer #1: I commend the authors for performing this interesting study which provides relevant and up to date information to the reader on the quality of life and cost of glaucoma care in a developing Sub Saharan African Country. however the following queries will need to be answered:

When was this study conducted ?

What was the inclusion criteria for enrolled participants, are there any exclusion criteria?

Table 1...... data for males should be included in the table

What in the authors opinion represent the salient limitations if any to their study?

The typographical errors in the manuscript should be corrected....Line 39 and 362

Reviewer #2: The authors investigate perceptions and local protocols of glaucoma treatment in Tansania during a local consensus meeting. Secondly they administered QoL questionnaires to a cohort of glaucoma patients.

Until now glaucoma blindness remains a mayor concern in rural Africa and so far there is no established gold standard for its treatment. The study questions are well outlined. The topic is relevant to practitioners in developing countries and beyond.

Focus group discussions revealed that most professionals still use drops as first line treatment; although limitations are known and obvious.

In their Discussions Authors should conclude that other treatment forms need to be implemented.

What about further surgical training for the midlevel cadre (AMO-O and ONOs)? Or shall we aim on building more specialized surgical glaucoma units?

I recommend to add some few extra lines on conclusions and recommendations.

6. PLOS authors have the option to publish the peer review history of their article (what does this mean?). If published, this will include your full peer review and any attached files.

Reviewer #1: Yes: Uche Nkechinyere J

Reviewer #2: No

---

## [Author Response · Author response to Decision Letter 0]

14 Apr 2020

Dear Prof. Peters

Re The cost and quality of life impact of glaucoma in Tanzania: an observational study

Many thanks for your review of our paper. We are pleased to submit a revised version for consideration for publication. Our response to the comments is given below.

Journal requirements

All formatting detail has been revised as requested. Other specific points relating to Journal Requirements are:

3 a We have no ethical constraints exist for publication of anonymized data and have included the data-set in supplementary materials since we are advocates of data sharing where at all feasible. We would be grateful if you could update our Data Availability statement accordingly.

4 The map in Figure 1 is sourced from Wikipedia. Wikipedia explicitly states the material has been released into the public domain. The citation for this has been included in the figure caption.

The map is taken from Wikipedia file: Tanzania Districts.png. (2014, November 25). Wikimedia Commons, the free media repository. Retrieved 21.12, April 13, 2020 from https://commons.wiki,media.org/w/index.php?title=File:Tanzania_District.png&oldid=140523089

5 A revised financial disclosure reads as follows.

Amended Funding Statement:

Dr Andrew F. Smith is a consultant to Medmetrics Inc and was a paid health economics consultant on this project which was fully sponsored by The British Council for the Prevention of Blindness (BCPB). Dr Smith was involved in the overall study design, data analysis and manuscript preparation phases of the entire project. 

The funder (BCPB) provided support in the form of salaries for authors [AFS] but did not have any additional role in the study design, data collection and analysis, decision to publish, or preparation of the manuscript. The specific roles of these authors are articulated in the ‘author contributions’ section.

Competing Interests Statement:

Dr Andrew F. Smith is a consultant to Medmetrics Inc and was a paid health economics consultant on this project. This does not alter our adherence to PLOS ONE policies on sharing data and materials.

No other authors have any competing interests to declare.

Review Comments to the Author

We are grateful to our reviewers for their time and expertise in assessing our manuscript. Our responses to their specific points are given below.

Reviewer #1: 

When was this study conducted ? 

What was the inclusion criteria for enrolled participants, are there any exclusion criteria?

The consensus meeting was conducted in 2017. This has been added to the text.

The patient questionnaires were completed in 2018. Convenience sampling was used in glaucoma clinics. Exclusion criteria were non-consent to participation and inability to respond to the questionnaires. Both facts have been added to the text.

Table 1…... data for males should be included in the table

Since gender is binary the proportion of males is a given once the proportion of females is stated. It is therefore not included for economy of space.

What in the authors opinion represent the salient limitations if any to their study?

We have discussed these in our discussion but agree a summary is beneficial for clarity hence have added the following to our conclusion:

‘In conclusion this study has the weaknesses discussed above of convenience sampling in an urban private clinic and use of subjective patient responses prone to lack of internal consistency. None-the-less the…..’

---

## [Editor Report · Decision Letter 1]

22 Apr 2020

The cost and quality of life impact of glaucoma in Tanzania: an observational study

PONE-D-19-34308R1

Dear Dr. Murdoch,

We are pleased to inform you that your manuscript has been judged scientifically suitable for publication and will be formally accepted for publication once it complies with all outstanding technical requirements.

With kind regards,

Remco PH Peters, MD, PhD, DLSHTM

Academic Editor

PLOS ONE
---

## [Editor Report · Acceptance letter]

14 May 2020

PONE-D-19-34308R1 

The cost and quality of life impact of glaucoma in Tanzania: an observational study 

Dear Dr. Murdoch:

I am pleased to inform you that your manuscript has been deemed suitable for publication in PLOS ONE. Congratulations! Your manuscript is now with our production department. 

With kind regards,

on behalf of

Prof Remco PH Peters 

Academic Editor

PLOS ONE